# Detection of Fast-Changing Intra-seasonal Vegetation Dynamics of Drylands Using Solar-Induced Chlorophyll Fluorescence (SIF)

Jiaming Wen[1,2†*], Giulia Tagliabue[3†], Micol Rossini[3], Francesco Pietro Fava[4], Cinzia Panigada[3], Lutz Merbold[5], Sonja Leitner[6], Ying Sun[1*]

[1]School of Integrative Plant Science, Soil and Crop Sciences Section, Cornell University, Ithaca, NY, USA.
[2]Department of Global Ecology, Carnegie Institution for Science, Stanford, CA, USA.
[3]Remote Sensing of Environmental Dynamics Laboratory, Department of Environmental and Earth Sciences (DISAT), University of Milano - Bicocca, Milano, Italy.
[4]Department of Environmental Science and Policy, Università degli Studi di Milano, Milano, Italy.
[5]Integrative Agroecology Group, Research Division Agroecology and Environment, Agroscope, Reckenholzstr. 191, 8046 Zurich, Switzerland.
[6]International Livestock Research Institute, Mazingira Centre, P.O. Box 30709, 00100 Nairobi, Kenya.

*Correspondence to*: Jiaming Wen (jwen@carnegiescience.edu), Ying Sun (ys776@cornell.edu)

†Jiaming Wen and Giulia Tagliabue contributed equally to this work.

**Abstract.** Dryland ecosystems are the habitat supporting two billion people on the Earth planet and strongly impact the global terrestrial carbon sink. Vegetation growth in drylands is mainly controlled by water availability with strong intra-seasonal variability. Timely availability of information at such scales (e.g., from days to weeks) is essential for early warning of potential catastrophic impacts of emerging climate extremes on crops and natural vegetation. However, the large-scale monitoring of intra-seasonal vegetation dynamics has been very challenging for drylands. Satellite solar-induced chlorophyll fluorescence (SIF) has emerged as a promising tool to characterize the spatiotemporal dynamics of photosynthetic carbon uptake and has the potential to detect intra-seasonal vegetation growth dynamics. Yet, few studies have evaluated its capability for detecting fast-changing intra-seasonal vegetation dynamics and advantages over traditional, vegetation indices (VIs)-based approaches in drylands. To fill this knowledge gap, this study utilized the vast dryland ecosystems in the Horn of Africa (HoA) as a testbed, to characterize their intra-seasonal dynamics inferred from satellite SIF. HoA is an ideal testbed because its dryland ecosystems have highly dynamic responses to short term environmental changes. The satellite data based analysis was corroborated with a unique *in-situ* SIF dataset collected in Kenya - so far, the only ground SIF time series collected in the continent of Africa. We found that SIF from TROPOspheric Monitoring Instrument (TROPOMI) with daily revisit frequency identified highly dynamic week-to-week variations in both shrublands and grasslands; such rapid-changing vegetation dynamics corresponded to the up- and down- regulation by the fluctuations of environmental variables (e.g., air temperature, vapor pressure deficit, soil moisture). However, neither reconstructed SIF products nor near-infrared reflectance of terrestrial vegetation (NIRv) from Moderate Resolution Imaging Spectroradiometer (MODIS), which is widely used in literature, was able to capture such fast-changing intra-seasonal variations. The same findings hold at the site scale, where we found only TROPOMI SIF revealed two separate within-season growth cycles in response to extreme soil

moisture and rainfall amount and duration, consistent with *in-situ* SIF measurements. This study generates novel insights on the monitoring of dryland vegetation dynamics and evaluation of their climate sensitivities, enabling development of predictive and scalable understanding of how dryland ecosystems may respond to future climate change and informing future design of effective vegetation monitoring systems for dryland vegetation.

## 1 Introduction

Drylands account for about 41% of the total terrestrial land surface and play a critical role in maintaining ecological functions and services, regulating global carbon cycles as well as contributing to socio-economic well-being (Prăvălie 2016; Poulter et al. 2014; Ahlström et al. 2015; Piao et al. 2020; Yao et al. 2020). In particular, drylands have been expanding globally in the recent decades (Lian et al., 2021) and are projected to continue expanding in the future (Huang et al., 2015). Therefore, it is of critical importance to understand how dryland ecosystems respond to the ongoing and future climate change for the sake of human welfares (Huang et al. 2017; Smith et al., 2019; Zhang et al. 2020a, 2022; Wang et al., 2022a). Vegetation growth in drylands is mainly controlled by water availability with strong intra-seasonal variability. Monitoring vegetation dynamics at the intra-seasonal scale (*e.g.*, from days to weeks) is critical for understanding climate impacts on carbon dynamics, detecting plant early stress and informing climate risk management (Otkin et al., 2018; Qing et al., 2022; Gerhards et al., 2019), as dryland ecosystems exhibit hyper-complex and rapid physiological/phenological dynamics at short time scales (Adams et al., 2021; Wang et al., 2022a). To do this, timely availability of information at such scales is crucial. However, for multiple reasons, intra-seasonal dynamics can be more challenging to monitor than trends at longer time scales such as inter-seasonal or inter-annual variations. First, the former characterizes variations that are mainly driven by changes in vegetation function (i.e., leaf physiology, such as photosystem redox states, nonphotochemical quenching, electron transport rate, etc., all of which affect the efficiency of light use) (Gu et al., 2019; Han et al., 2022; Sun et al., 2023a), while the latter characterizes variations that are largely driven by changes in vegetation structure (e.g., leaf area, leaf angle, or pigment content, all of which affect light absorption and scattering) (Li et al., 2024). Second, the time window is shorter for the former than for the latter, with less observation sampling for accurate depiction of temporal dynamics. Consequently, detecting fast-changing intra-seasonal vegetation dynamics for early warning purposes requires high-frequency observations that are sensitive to dryland functional changes.

Greenness-based vegetation indices (VIs), such as Normalized Difference Vegetation Index (Tucker et al., 1979) and near-infrared reflectance of terrestrial vegetation (NIRv, Badgley et al., 2017), from Earth Observation (EO) satellites have been used for vegetation monitoring for decades (Qu et al., 2019; Lawal et al., 2021; Ouma et al., 2022; Fava et al., 2021). For example, NIRvP, the product of NIRv and photosynthetically active radiation (PAR), was found to be a robust structural proxy for photosynthesis (Dechant et al., 2022). In the recent decade, solar-induced chlorophyll fluorescence (SIF) has emerged as a promising proxy for inferring photosynthetic dynamics from canopy to global scales (Porcar-Castell et al.,

2014; Sun et al., 2023a, 2023b). SIF has unique mechanistic advantages as it is emitted from the core of the photosynthetic machinery and therefore contains additional functional information (e.g., light use efficiency) beyond structural information (e.g., light absorption) that is usually carried by VIs. In addition, since SIF signal comes only from active vegetation, it is less susceptible to the brightness of soil background, unlike reflectance-based VIs (Huete et al., 2002). These characteristics make SIF a unique observational signal for inferring photosynthetic dynamics for dryland ecosystems. For example, SIF has demonstrated a superior capability in accurately depicting dryland ecosystem phenology (Wang et al., 2019) as well as capturing seasonal variations (Wang et al., 2022c) and interannual variations (Smith et al., 2018) of in situ gross primary production (GPP). Furthermore, it has facilitated many applications in drought detection and ecosystem restoration in drylands (Robinson et al., 2019; Mengistu et al. 2021, Constenla-Villoslada et al., 2022). However, most of such evaluation were conducted at the seasonal scale or beyond, and very few have been focused on short time scales, e.g., intra-seasonal. We hypothesize that SIF may present more complex intra-seasonal dynamics due to functional changes in response to short-term environmental fluctuations, while NIRv remains relatively constant as there are minimal structural changes at a temporal scale of several days to weeks especially during the peak growing season.

To test this hypothesis, we utilized dryland ecosystems in the Horn of Africa (HoA, Fig. 1a) as a testbed to evaluate the capacity of satellite SIF and NIRv in capturing the intra-seasonal vegetation dynamics of drylands. The HoA has experienced frequent droughts and excessive rainfall (Williams et al., 2012; Lyon and Dewitt, 2012; Funk et al., 2015; Ngoma et al., 2021) and suffered strong vulnerability to climate change (IPCC Working Group II, Chapter 9, 2022). The highly dynamic vegetation growth in response to volatile environmental conditions puts millions of pastoralists and smallholder farmers at risk (Matanó et al., 2022) and exacerbates the persistent food insecurity challenges in this region (Pricope et al. 2013; Beal et al. 2023), calling for accurate and prompt vegetation monitoring and early warning systems (Merbold et al. 2021). In particular, in this study we focused on the period from October 2019 to February 2020, when excessive rainfall occurred in the HoA (Fig. 1e), leading to anomalous vegetation dynamics that are challenging to be accurately depicted by satellite measurements. We employed multiple high-temporal-resolution satellite SIF products, including original SIF retrievals from TROPOspheric Monitoring Instrument (TROPOMI, with unprecedented daily revisit frequency for satellite SIF retrieval, Köhler et al., 2018; Guanter et al., 2021) and several machine-learning reconstructed SIF products (at a temporal resolution from 4-day to 16-day), and NIRv from Moderate Resolution Imaging Spectroradiometer (MODIS) (at daily resolution), along with a unique ground SIF dataset measured at an environmental research infrastructure site located in Kenya - so far, the only in situ SIF time series reported in the continent of Africa.

This paper is structured as follows: Sect. 2 introduces the region of interest and datasets employed in this study. Sect. 3 evaluates different satellite SIF products with in-situ SIF time series (Sect. 3.1) and investigates the intra-seasonal vegetation dynamics under excessive precipitation at site and regional levels (in Sect. 3.2, Sect. 3.3, respectively). Sect. 4 discusses possible reasons and implications for discrepancies among different datasets. Sect. 5 summarizes the conclusions.

## 2 Study region and datasets

### 2.1 The HoA drylands

The HoA region is located in eastern Africa, including Somalia, Ethiopia, Kenya, Eritrea, and Djibouti, with most area covered by drylands (Fig. 1a). From the eastern coast to inner highlands, there is a general gradient of increasing water availability (Fig. 1d), which drives a land cover shift from barren areas, to shrublands, and to grasslands (Fig. 1b), with a corresponding variation in vegetation greenness (Fig. 1c). The HoA is signatured by a short rainy season (SR, usually from October to the following January, with variations depending on the location) and a long rainy season (LR, usually from March to June), with two dry seasons in between (Fig. 1h). Vegetation thrives during the rainy seasons and wanes during the dry seasons (Fig. 1f, 1g). During the short rainy season in 2019 (i.e., October 2019 – January 2020), the HoA experienced anomalously high precipitation compared to normal years (Fig. 1e). 50% of the total area had precipitation two standard deviations (> 2 σ) higher than normal years, mostly in central and southern HoA drylands, and another 39% of the area had precipitation one to two standard deviations (1 – 2 σ) higher than normal years. In this study, we selected three sub-domains of interest to investigate the intra-seasonal vegetation dynamics under excessive precipitation: Region 1 (including eastern Ethiopia and central Somalia, dominated by shrublands), Region 2 (including southern Somalia, dominated by grasslands), and Region 3 (i.e., Kenya, dominated by grasslands) (Fig. 1b). These three sub-domains were selected, because within each sub-domain, (1) the land cover type is relatively homogeneous; and (2) the precipitation pattern and vegetation response are relatively consistent (Sect. 3.3).

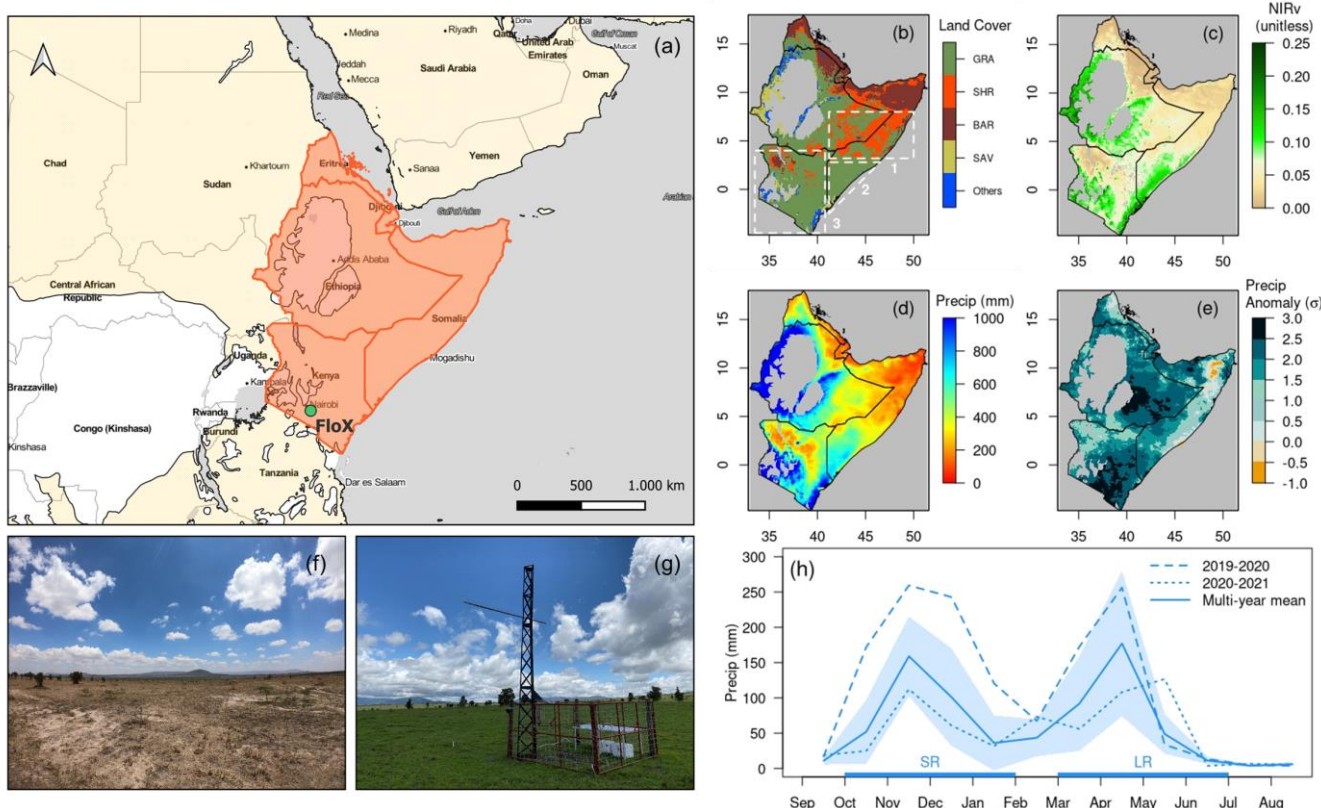

Figure 1: (a) Spatial extent of the HoA (orange) and drylands (light yellow) (defined as areas where the ratio of precipitation and potential evapotranspiration, i.e., aridity index (AI), is less than 0.65, Sorensen et al., 2007). The location of the FloX tower is marked as a green dot. (b) MODIS land cover map (Friedl and Sulla-Menashe 2022) of the HoA in 2019. The land cover categories are: grasslands (GRA), shrublands (SHR), barren areas (BAR), savannas (SAV), and others. The three white dashed squares mark the three sub-domains of interest in our regional analysis. (c) – (d) Spatial maps of multi-year mean (2011-2020) of near-infrared reflectance of vegetation (NIRv) and annual precipitation (Precip), respectively. (e) Spatial map of the standardized anomaly of precipitation during the SR season in 2019 (from October 2019 to January 2020) relative to the multi-year SR mean (2011-2020), in the unit of standard deviation σ. (f) – (g) Pictures of the grasslands at Kapiti where the FloX tower is located, captured before (September 28th, 2019) and during (October 26th, 2019) SR, respectively. (h) Time series of precipitation at Kapiti during 2019-2020 (dashed) and 2020-2021 (dotted), when in situ SIF was collected, compared to the multi-year mean (2011-2020, solid). The shade denotes one standard deviation of monthly precipitation during 2011 and 2020. The lengths of SR and LR seasons are marked on the x-axis in light blue.

## 2.2 Description of in situ SIF collection: site characteristics, instrumentation, and SIF retrieval algorithms

**Site description**: The Kapiti Research Station and Wildlife Conservancy (from now called Kapiti for simplicity) is a research facility owned and managed by the International Livestock Research Institute (ILRI) located in Machakos county of southern Kenya (Fig. 1a). Kapiti, largely characterized by flat or gently sloped topography, covers approximately 13,000 ha

and is located at about 1,650 m above mean sea level (Dowling et al. 2022, Carbonell et al. 2021). Kapiti is dominated by semi-arid vegetation, including grasses, shrubs and isolated trees (Fig. 1f, 1g). The climate is semi-arid with an average annual precipitation of approximately 500 mm distributed among two main rainy seasons (Fig. 1h). However, the mean annual precipitation and the seasonal distribution of precipitation are highly variable, with frequent droughts or excess rain episodes.

**In situ instrument**: In situ SIF data used in this study were collected from a tower positioned in a flat area of the Kapiti site dominated by open grasslands (1.6144°S, 37.1338°E, Fig. 1a). SIF was measured using the fluorescence box (FloX, JB Hyperspectral Devices GmbH, Germany), an automatic hyperspectral device for the continuous observation of SIF and reflectance. The FloX system consists of two internal spectrometers (Ocean Insight, USA) contained in a temperature-controlled case. The first spectrometer (i.e., QEPro) covers the spectral range 650-800 nm with a full width at half maximum (FWHM) of 0.3 nm and is specifically designed for the retrieval of SIF. The second spectrometer (i.e., FLAME) covers a broader spectral range (400-950 nm) with a FWHM of 1.5 nm and is intended for the observation of reflectance. Each spectrometer measures the downwelling irradiance with an up-looking cosine optic, as well as the upwelling radiance with a down-looking bare optical fiber (25° field of view). The down-looking fibers were placed nadir-looking at a height of 4.5 m above the ground, which corresponds to a footprint of ca. 1.9 m diameter. The system was installed at the Kapiti research site on September 25th, 2019, and has been measuring continuously until August 31st, 2021.

**Processing of in situ SIF**: The FloX raw data were processed using a dedicated R script (R Core Team, 2022) developed by the manufacturer (v. 20.7). The processing included the conversion from raw data to radiance using the calibration files of the spectrometers, the retrieval of SIF, the calculation of apparent reflectance and the computation of quality flags. SIF was retrieved at the $O_2$-A absorption band (i.e., 760 nm) using both the improved Fraunhofer Line Depth method (iFLD) (Alonso et al., 2008) and the Spectral Fitting Method (SFM) (Cogliati et al., 2015), denoted as $FloX_{iFLD}$ SIF and $FloX_{SFM}$ SIF. For the iFLD, we used the bands at 756.04 nm and 760.05 nm outside and within the absorption band, respectively, while for the SFM we used a fitting window of 750.12-779.90 nm. A multiplicative wavelength conversion factor of 1.72 from Yang et al. (2015) was applied to the retrieved SIF values to allow comparison with satellite SIF datasets derived at 740 nm. The data from both the QEPro and FLAME spectrometers were then filtered to discard low-quality measurements. The filtering criteria were defined as follows: (a) solar zenith angle (SZA) less than 70°; (b) incoming solar radiation variation (i.e., percent difference between the irradiance measurement before and after each target measurement) less than 1%; (c) dynamic range of the spectrometer between 60% and 90%; (d) clearness index (i.e., the ratio between actual and potential solar irradiance, Chang et al. 2020) between 0.9 and 1.1.

### 2.3 Satellite vegetation datasets

**TROPOMI SIF:** The TROPOMI instrument onboard Sentinel-5 Precursor (S-5P) satellite was launched in October 2017, with an equatorial overpass time at 13:30 local solar time. It has a spatial resolution of $3.5 \times 7.5$ km$^2$ ($3.5 \times 5.5$ km$^2$ since August 2019), with a wide swath (~2600 km) that enables daily global coverage (Köhler et al., 2018; Guanter et al., 2021).

There are three TROPOMI SIF datasets available, one provided by the California Institute of Technology (Caltech) with fitting window 743-758 nm (Köhler et al., 2018), the other two by the European Space Agency (ESA) with fitting windows 735-758 nm and 743-758 nm (Guanter et al., 2021). All datasets are retrieved using the singular value decomposition (SVD) approach. We employed two different thresholds of cloud fraction (CF) for SIF intercomparison and intra-seasonal vegetation dynamics analysis, following Guanter et al. (2021): we selected Level 2 SIF retrievals with CF less than 0.2 when we compared TROPOMI SIF with in situ SIF (Sect. 3.1) to minimize the cloud influence on the retrieved SIF; we applied a less stricter rule (CF less than 0.8) when we used TROPOMI SIF to evaluate vegetation dynamics (Sect. 3.2 and 3.3), to enable a good temporal sampling. Level 2 SIF retrievals with SZA larger than 70° were excluded. All selected level 2 SIF retrievals were first converted to daily corrected SIF based on SZA (Frankenberg et al., 2011) and then re-gridded to a 0.15° pixel using a gridding tool (https://github.com/cfranken/gridding). 0.15° was selected to include enough soundings for spatial aggregation to reduce measurement noise while maintaining overall representativeness of the area around the tower (Fig. S1).

**Reconstructed SIF products**: CSIF (version 2, Zhang et al., 2018), GOSIF (version 2, Li and Xiao, 2019) and SIF_oco2_005 (updated version based on OCO-2 v10r retrievals, Yu et al., 2019) are reconstructed based on SIF retrievals from OCO-2. OCO-2, launched in 2014, provides SIF retrievals at a resolution of $1.3 \times 2.25$ km$^2$ with a 16-day revisit cycle and an equatorial overpass time at 13:30 local solar time (Sun et al. 2018). One of the limitations of OCO-2 SIF retrievals is the incomplete global coverage, with large spatial gaps between satellite tracks. The overall strategy for generating these reconstructed SIF products is similar: (1) establishing statistical relationships between available OCO-2 SIF measurements and ancillary variables (e.g., surface reflectance, vegetation indices, meteorological forcings) using machine learning algorithms (e.g., neural networks, cubic regression tree model); (2) applying the relationship on ancillary variables with global coverage to fill the gaps where OCO-2 retrievals are not available. These three products differ in the choice of machine learning approaches and ancillary variables that were used to generate them. They are provided at a spatial resolution of 0.05° and a temporal resolution of 4-day, 8-day, and 16-day, respectively. A wavelength correction factor of 1.69 was multiplied to the three OCO-2 based SIF products (evaluated at 757 nm) to match with TROPOMI SIF (evaluated at 740 nm). In addition, we also employed RTSIF, a recent reconstructed SIF dataset based on TROPOMI SIF (Chen et al., 2022). As TROPOMI SIF is only available since 2018, Chen et al. (2022) similarly utilized a machine learning algorithm and ancillary datasets to reconstruct a long-term SIF record during 2001-2020, at 0.05° and 8-day resolution.

**SIF yield:** SIF yield carries information on plant physiological/functional variations in response to environmental changes (Sun et al., 2015; Yoshida et al. 2015; Yang et al., 2015; Miao et al., 2018; Magney et al., 2019; Sun et al., 2023a). In this study, to tease out the plant functional variations from structural variations contained in the remotely sensed SIF signal, we derived SIF yield = SIF / PAR / NIRv, following Dechant et al., 2020.

**MODIS NIRv**: NIRv used in this study was calculated from MODIS MCD43A4 (Version 6.1) Nadir Bidirectional Reflectance Distribution Function (BRDF)-Adjusted Reflectance (NBAR) dataset (Schaaf and Wang, 2021), provided at

daily and 500 m resolution. To maintain a good sample size for vegetation dynamics analysis, we kept the data with quality flags as 0 (full BRDF inversions) or 1 (magnitude inversion), following Wang et al. (2018).

## 2.4 Climate variables

Precipitation data were obtained from Climate Hazards group Infrared Precipitation with Stations (CHIRPS, version 2.0) (Funk et al., 2015). CHIRPS covers 50°S - 50°N from 1981 to present at 0.05° and daily resolution and is generated by incorporating Cold Cloud Duration (CCD) from satellite observations and ground data from rain gauges (Funk et al., 2015). CHIRPS precipitation estimates have shown a great agreement with ground data in Africa (Dinku et al., 2018; Ayehu et al., 2018; Ageet et al., 2021; Ngoma et al., 2021).

Soil moisture (SM) was from ESA-CCI (v06.1) by the European Space Agency (ESA) Climate Change Initiative (CCI) program, offered at 0.25°, daily resolution from 1978 to 2020 (Preimesberger et al., 2021). It was generated by harmonizing the soil moisture estimates (typically at a depth of 0-5 cm) from multiple active and passive satellite microwave sensors (Dorigo et al., 2017; Gruber et al., 2019). In this study, we employed an updated version from Preimesberger et al. (2021).

Air temperature (Tair), water vapor pressure deficit (VPD) and PAR, at the OCO-2 and TROPOMI nominal overpass time at the equator (i.e., 13:30 local solar time), were extracted from the Global Modeling and Assimilation Office (GMAO) Modern-Era Retrospective analysis for Research and Applications, Version 2 (MERRA-2) reanalysis (hourly, lon 0.625°× lat 0.5°) (GMAO, 2015a, 2015b).

## 2.5 Spatial and temporal matching criteria

We employed multiple spatial and temporal matching criteria for the intercomparison among different SIF datasets (Sect. 3.1) and for the analysis of intra-seasonal vegetation dynamics (Sect. 3.2 and 3.3). The principle is as follows: for the SIF intercomparison against in situ SIF, we aimed to ensure the best spatial/temporal consistency between in situ SIF and each satellite SIF dataset to be evaluated; for the intra-seasonal analysis, we attempted to ensure the spatial/temporal consistency among all the datasets (including SIF and other ancillary variables) so that all the variables refer to the same spatial domains and time intervals.

### 2.5.1 Spatial and temporal matching criteria for SIF intercomparison

**Spatial matching:** for comparison with in situ SIF measurements, TROPOMI was regridded to 0.15° pixel (as explained in Sect. 2.3) centered at the tower location. For the reconstructed SIF products, we extracted the value of the 0.05° pixel where the tower is located to minimize the difference in spatial scales.

**Temporal matching:** For the paired comparison between in situ SIF and TROPOMI, we selected the quality-filtered in situ SIF observations (Sect. 2.2) collected within a time window of ±30 minutes with respect to the overpass time of each TROPOMI observation. The selected measurements were averaged after applying the daily-correction factor based on the

SZA, which was also applied to the TROPOMI SIF. The TROPOMI observations for which no in situ SIF observations were available in the ±30 minutes time window were discarded. In total, 64 data pairs were used for comparison.

For the paired comparison between in situ SIF and the reconstructed SIF products, we extracted the quality-filtered in-situ SIF within a ±30 minutes time window centered at the OCO-2 and TROPOMI nominal overpass time at the equator (i.e., 13:30 local solar time), applied the daily correction factor and then averaged the measurements across 4-, 8- or 16-day periods to match with the temporal resolution of the reconstructed products. In total, 67, 84, 40 and 55 data pairs were used for comparison for CSIF, GOSIF, SIF_oco2_005 and RTSIF, respectively.

**2.5.2 Spatial and temporal matching criteria for intra-seasonal analysis**

To ensure consistency among all datasets, we aggregated or resampled all datasets (e.g., in-situ SIF, TROPOMI SIF, MODIS NIRv, and climate variables) to the same 0.15° and 8-day resolutions. The 0.15° pixels were set so that the boundary of the 0.15° pixel around the tower was aligned with the 3 × 3 0.05° pixels of GOSIF and RTSIF closest to the tower. Therefore, this is slightly different from the 0.15° pixel for TROPOMI SIF described in Sect. 2.5.1. The 8-day resolution was selected,

(1) to reduce the measurement noise of TROPOMI SIF and in situ SIF while preserving the fine-scale intra-seasonal temporal variations, and (2) to match the coarser temporal resolution of GOSIF and RTSIF. For the analysis of intra-seasonal dynamics at Kapiti (Sect. 3.1), we selected the quality-filtered in situ SIF observations (Sect. 2.2) collected within a time window of ±30 minutes with respect to the overpass time of TROPOMI and applied a daily-correction factor based on the SZA to convert them into daily values. The daily values were then aggregated to the same 8-day intervals.

**3 Results**

**3.1 Evaluation of satellite SIF datasets with in-situ SIF**

Leveraging the in situ SIF time series at Kapiti, here we evaluated the fidelity of various satellite-based SIF datasets during two consecutive years (i.e., from September 2019 to August 2021) when in situ SIF was collected (Figs. 2, S2). In situ SIF showed strong inter-annual variations, with a much stronger signal in the first year compared to the second year, driven by

255 the difference in precipitation between the two years (Fig. 1h). It also exhibited pronounced intra-annual variations such as growth peaks during SR seasons (e.g., November 2019 - January 2020, December 2020), LR seasons (e.g., May 2020, June 2021), and a dry season with intermittent precipitation (February - March 2021, Fig. 1h) (Fig. 2a). The satellite-based SIF datasets showed different degrees of consistency with in situ SIF. The temporal dynamics of in situ SIF were well captured by TROPOMI SIF (Fig. 2a), showing high agreement against FloX SIF$_{iFLD}$ (r = 0.71-0.83, Fig. 2c-2e), with slightly reduced

agreement against FloX SIF$_{SFM}$ (r = 0.64-0.76, Fig. S2c-S2e). Instead, the reconstructed SIF products (i.e., CSIF, GOSIF, SIF_oco2_005, RTSIF), although highly consistent among each other, showed a greater discrepancy with in situ SIF compared to TROPOMI (Fig. 2b). The reconstructed SIF products showed less frequent intra-seasonal variations, and their magnitudes of variations are sometimes inaccurate (e.g., the drop in December 2019 and the peak in February - March

2021), leading to lower correlation against FloX SIF$_{iFLD}$ compared to TROPOMI (r = 0.58-0.62, Fig. 2f-2i). Their correlation against FloX SIF$_{SFM}$ became not significant (Fig. S2f-S2i), likely because the SFM approach with a wide fitting window is more sensitive to atmospheric contamination (Chang et al., 2020). This is probably magnified when the data are aggregated over time windows of several days, such as in the comparison against the reconstructed SIF products. Note that some negative values appear in FloX SIF and TROPOMI SIF especially during the time periods with weak SIF signals, as a result of measurement and retrieval noise (Guanter et al., 2021). These negative values are retained in the evaluation to avoid an artificial positive bias in spatial and temporal aggregation.

In the following analysis of intra-seasonal vegetation dynamics, we only selected a subset of SIF datasets. We selected: FloX SIF$_{iFLD}$ because of its lower data noise compared to FloX SIF$_{SFM}$; TROPOMI SIF from ESA (fitting window 743-758 nm) because of its higher consistency with in situ SIF compared to the other two TROPOMI SIF datasets (Figs. 2d, S2d); GOSIF as a representative of the three OCO-2 based reconstructed SIF products given the overall consistency among them; RTSIF as it is a TROPOMI-based reconstructed SIF product.

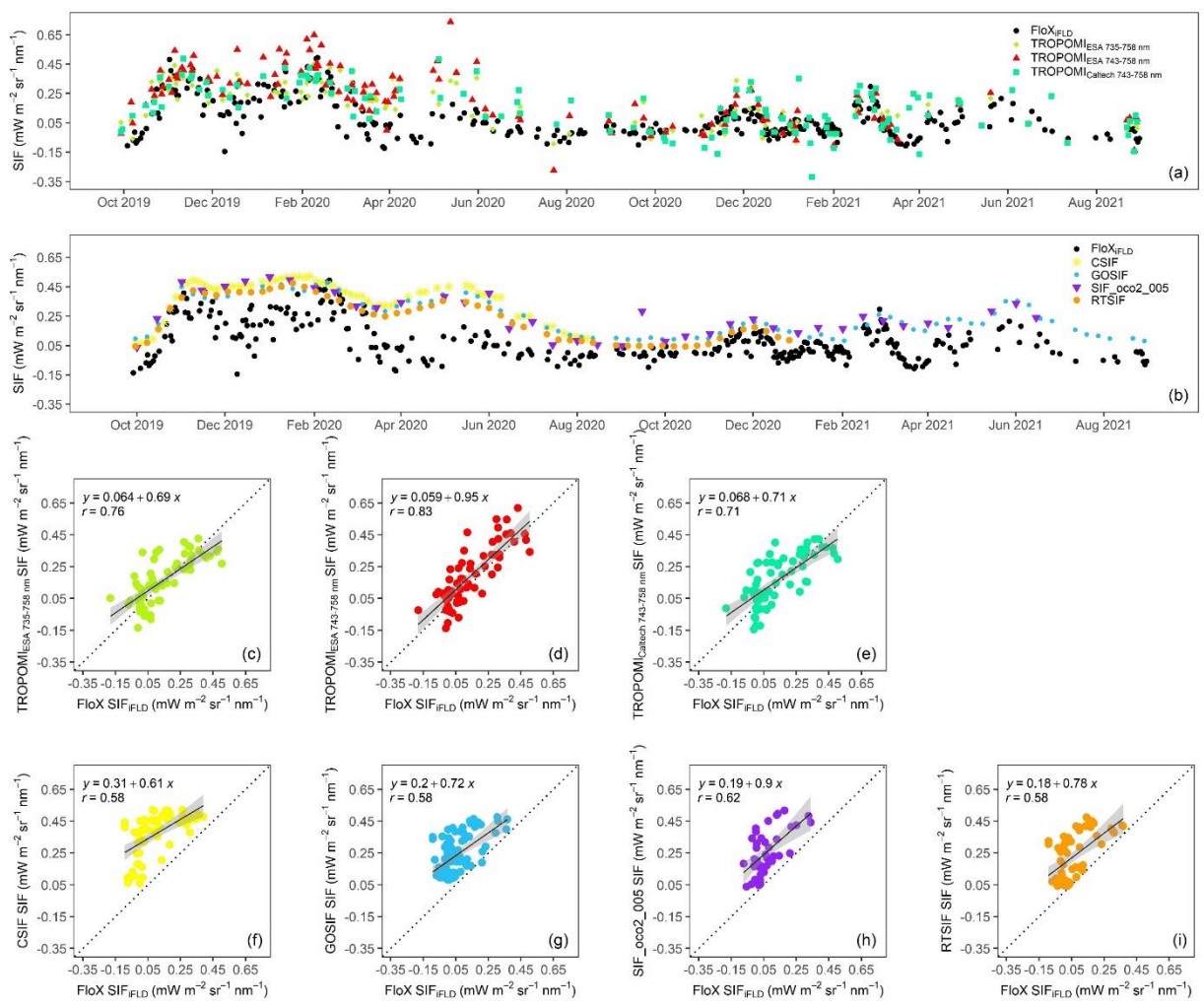

**Figure 2: (a), (b) Time series of FloX SIF$_{iFLD}$ and satellite SIF at 740 nm from October 2019 to September 2021. (c) - (e) Scatterplots between FloX SIF$_{iFLD}$ and TROPOMI SIF from ESA (fitting window 735-758 nm and 743-758 nm) and Caltech, respectively. (f) - (i) Scatterplots between FloX SIF$_{iFLD}$ and CSIF, GOSIF, SIF_oco2_005, RTSIF, respectively. All SIF values are daily corrected. The dotted line marks the 1:1 line.**

### 3.2 Intra-seasonal dynamics at Kapiti

We evaluated the capability of satellite SIF and NIRv in characterizing the intra-seasonal vegetation dynamics at Kapiti from October 2019 to February 2020 (i.e., the SR season and the subsequent dry season) (Fig. 3), where/when in situ data (including SIF) are available to help verify and interpret the intra-seasonal dynamics. This period was chosen because excessive precipitation occurred during this SR season (i.e., 799 mm relative to the 2011-2020 average 343 ± 170 mm, Fig. 1h), leading to complex vegetation dynamics that can be challenging to be accurately characterized by satellite measurements. These challenges arise mainly from limited temporal frequency and/or spatial resolution of satellite data that

can easily miss fast-changing vegetation functions. Therefore, our chosen period is unique in evaluating the efficacy of
satellite measurements in capturing such complex dynamics.

We found that there was a rapid growth revealed in all SIF datasets in response to precipitation and soil moisture increase in October 2019 (Fig. 3a, 3b). NIRv showed a similar increase during this period. However, divergence among different SIF and NIRv datasets started to emerge in early November 2019 and persisted through February 2020. The reconstructed SIF products (i.e., RTSIF, GOSIF) and MODIS NIRv remained relatively stable from November 2019 to mid-January 2020
before a subsequent gradual decline. In contrast, TROPOMI SIF exhibited distinct dynamics during this period, with double peaks in mid-November 2019 and late January, and a sharp reduction (by 52% relative to the first peak) in between. This double-peak pattern in TROPOMI SIF held, regardless of sources of TROPOMI data, fitting windows used for SIF retrievals, or quality filtering criteria (e.g., SZA, CF, and retrieval error) (Fig. S3a-S3d). The double-peak pattern was not an artifact of variations in escape probability or sun-viewing geometry, but was a result of the true SIF emission (Fig. S3d).

In situ SIF confirmed these distinct intra-seasonal dynamics depicted by TROPOMI SIF, with similar magnitude (61%) and duration of the mid-season dip (Fig. 3b). As the product of NIRv and incoming PAR (i.e., NIRvP, Dechant et al., 2022) has been recently promoted as a strong proxy for photosynthesis, we further computed NIRvP with in situ NIRv and PAR. However, we found that it only accounted for a limited extent of mid-season reduction (22%, relative to the maximum in mid-November) (Fig. S4). This finding suggests that 1) suppression of PAR during the excessively rainy period was not the
cause of the observed SIF reduction, and that 2) NIRv itself is insufficient to timely capture the rapid and complex intra-seasonal dynamics.

To better demonstrate the intra-seasonal temporal dynamics, we further calculated the temporal change rate for each dataset (i.e., temporal changes between two consecutive time steps that are 8 days apart), to present the rate of temporal fluctuations (Fig. 3c). It was evident that both in situ SIF and TROPOMI SIF showed strong intra-seasonal variations, while the
reconstructed SIF products and MODIS NIRv presented minimal intra-seasonal variations.

What are the underlying processes driving such fast-changing intra-seasonal dynamics revealed by in situ and TROPOMI SIF? The strong mid-season reduction in SIF likely resulted from functional changes in vegetation photosynthetic activities, driven by grass phenology due to persistent rainfall (Reyer et al., 2013; Zeppel et al., 2014) (Fig. 3a). The onset of the herbaceous vegetation growth occurred in October 2019 triggered by abundant precipitation and soil moisture; the growth
peaked in early November 2019 as alluded by the Phenocam images collected at Kapiti (Fig. S5a). The grass progressed to the reproductive stage during early and mid December (Fig., S5b, Cheng et al., 2020; Zhang et al., 2023), resulting in a gradual decrease in the photosynthetic activity, possibly because of nutrient remobilization and carbohydrate sink limitation (Tejera-Nieves et al., 2023). At the same time, the persistence of soil moisture facilitated the onset of a new growth cycle, likely with a species composition shift (Muthoka et al., 2022; Shaw et al., 2022), which reached its second peak in early
February 2020 (Fig. S5c). However, such complex intra-seasonal dynamics cannot be captured by NIRv or the reconstructed SIF (thorough discussions in Sect. 4).

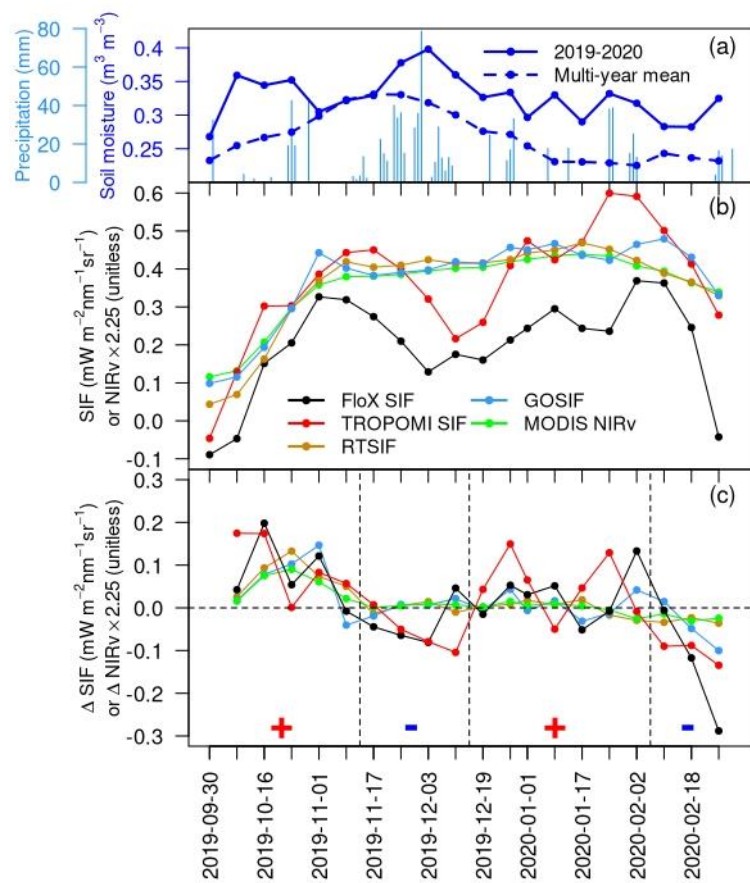

Figure 3: (a) Temporal variation of daily precipitation and 8-day average SM at Kapiti between October 2019 and February 2020. The multi-year average of SM during 2011-2020 is plotted as a blue dashed line for reference. (b) Temporal variation of vegetation signals at Kapiti from various SIF and NIRv datasets. A factor of 2.25 was multiplied to NIRv to match the magnitude range of SIF for visual clarity. (c) Temporal change rate of SIF (ΔSIF) or NIRv (ΔNIRv), calculated as the change of the current 8-day period relative to the previous 8-day period. The horizontal dashed line denotes no change in SIF or NIRv. The vertical dashed lines roughly divide the study period into four segments based on the sign of ΔSIF of in situ and TROPOMI SIF (mostly consistent with each other, as marked along the x axis). The x axis labels represent the starting date of each 8-day interval.

### 3.3 Intra-seasonal dynamics for the entire HoA drylands

Does the stronger sensitivity of TROPOMI SIF (compared to the reconstructed SIF and NIRv) in characterizing fast-changing intra-seasonal dynamics hold across HoA drylands, beyond the single site at Kapiti? To answer this question, we conducted in-depth regional analysis for the entire HoA drylands from October 2019 to February 2020 when excessive precipitation occurred in most of the region (Fig. 1e). Given the outstanding spatial heterogeneity of biome types, precipitation patterns and vegetation responses in the HoA, we selected three sub-domains for analysis (Fig. 1b), to ensure that within each sub-domain, 1) the land cover type is relatively homogeneous; and 2) the intra-seasonal variations of precipitation and subsequent vegetation growth were relatively consistent (Figs. S6, S7). For example, Region 1 and Region 2 (in central and southeastern HoA, dominated by shrublands and grasslands, respectively) started their rainy season in early

October, which stimulated fast vegetation growth. The vegetation activity peaked around early November and gradually decreased after December when there was little precipitation. In contrast, in Region 3 (in southern HoA, dominated by grasslands) precipitation occurred later (e.g., mainly during late October and early December). Correspondingly, the vegetation phenology was shifted with a peak around early December.

While all the satellite SIF and NIRv datasets well tracked the seasonal variations of the three sub-domains, we found that
TROPOMI SIF revealed more intra-seasonal variations during the growing seasons compared to the reconstructed SIF (i.e., RTSIF and GOSIF) and MODIS NIRv. In addition, TROPOMI SIF also showed higher values during the peak growing season and lower values during the dry season (i.e., February).

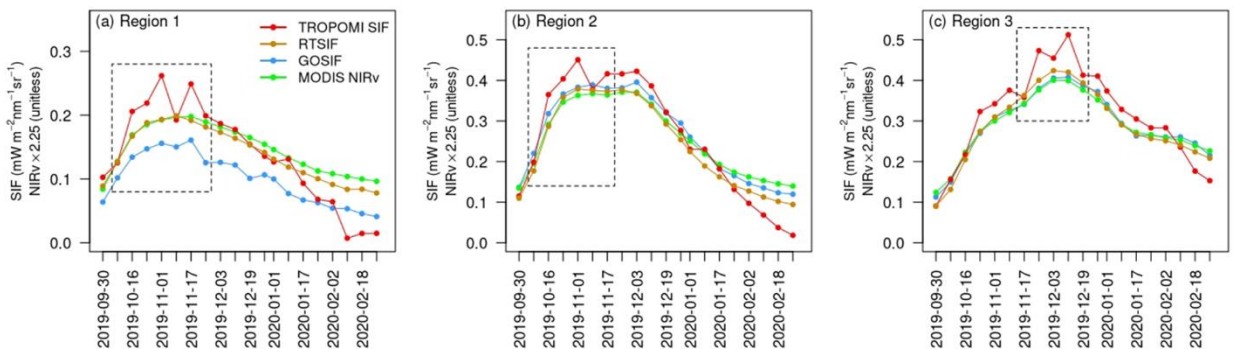

**Figure 4: Temporal variations of various SIF and NIRv datasets for the three sub-domains (Fig. 1b) in the HoA drylands from October 2019 to February 2020. A factor of 2.25 was multiplied to NIRv in order to match the magnitude range of SIF for visual clarity. The date labels represent the starting date of each 8-day interval. The dashed boxes mark the periods when TROPOMI SIF revealed strong intra-seasonal variations.**

To investigate the intra-seasonal variations revealed by TROPOMI SIF, we zoomed into a shorter time window for each of the sub-domains (i.e., dashed boxes in Fig. 4). For each time window, TROPOMI SIF showed a faster and stronger increase from a similar starting point, compared to the reconstructed SIF and MODIS NIRv. As a result, TROPOMI SIF showed a much stronger vegetation signal (i.e., higher values) during the peak growing season, e.g., November 1[st] in the central area of Region 1 (Fig. 5), November 1[st] in the coastal area of Region 2 (Fig. S8), November 25[th] in the central and southern area of
Region 3 (Fig. S10). After reaching a peak or close-to-peak value, TROPOMI SIF showed a decline during all three selected windows, e.g., a region-wide reduction for Region 1 and 2 on November 9[th], and a reduction in the central and southern area of Region 3 on December 3[rd]. These reductions in TROPOMI SIF were quickly recovered within a week. For Region 1 and Region 3, there was another subsequent regional-wide sharp reduction, on November 25[th] and December 19[th] respectively, before the vegetation activity gradually ceased. Figures. 6, S9, S11 depicted the temporal change rate of different SIF and
NIRv datasets. While TROPOMI revealed strong intra-seasonal variations during the peak growing season, the reconstructed SIF and MODIS NIRv remained nearly invariant.

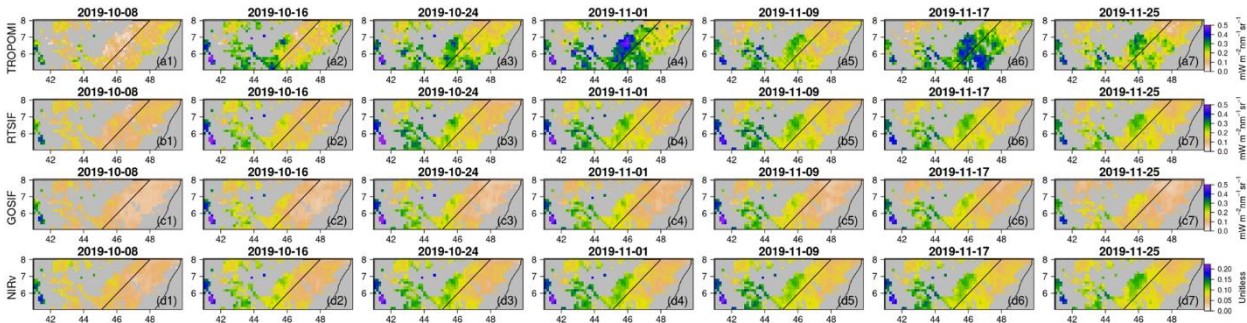

**Figure 5: Intra-seasonal variations of (a) TROPOMI SIF, (b) RTSIF, (c) GOSIF, (d) MODIS NIRv in the shrublands of Region 1 during October 8th and November 25th, 2019. The date labels represent the starting date of each 8-day period.**

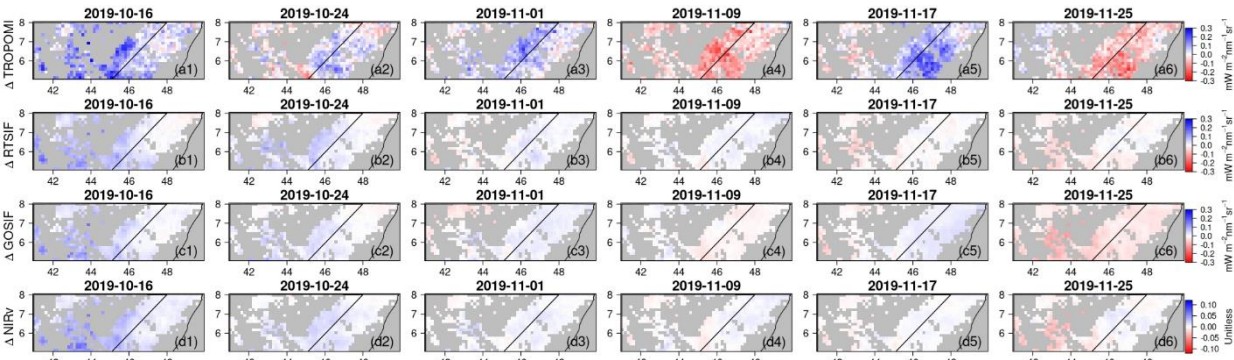

**Figure 6: Temporal change rate of (a) TROPOMI SIF, (b) RTSIF, (c) GOSIF, (d) MODIS NIRv in the shrublands of Region 1 during October 16th and November 25th, 2019. The date labels represent the starting date of each 8-day period.**

To identify drivers underlying the intra-seasonal variations observed in TROPOMI SIF, we further investigated meteorological variables from MERRA-2 and ESA-CCI SM (Fig. 7). We found that the reductions in TROPOMI SIF (e.g., Region 1 on November 9th and 25th, Region 2 on November 9th, and Region 3 on December 3rd and 19th) mostly coincided with increased Tair and VPD, and decreased SM. On the other hand, the subsequent recoveries (e.g., Region 1 and 2 on November 17th, and Region 3 on December 11th), all corresponded to decreased Tair and VPD, and increased SM. Such relationships between TROPOMI SIF and meteorological variations suggest that the intra-seasonal variations observed in TROPOMI SIF may represent the real vegetation status and are less likely artifacts of data noise. Again, the reconstructed SIF and MODIS NIRv, on the contrary, failed to capture such fast-changing intra-seasonal vegetation dynamics driven by environmental fluctuations (Fig. S12). With variations in PAR mostly showing opposite changes to variations in TROPOMI SIF (e.g., Fig. 7a), NIRvP could not capture such intra-seasonal variations either.

Furthermore, we found that the SIF yield calculated from TROPOMI SIF (i.e., SIF yield = SIF / PAR / NIRv, following Dechant et al., 2020) has an even higher consistency with the short-term fluctuations in Tair, VPD and SM (Fig. 7). This

further suggests that the intra-seasonal variations in TROPOMI SIF are largely driven by the functional changes regulated by environmental conditions. Interestingly, while TROPOMI SIF showed a slight increasing trend in Region 2 during October 16th and November 1st, TROPOMI SIF yield showed a large decreasing trend which corresponded to an increasing trend of Tair and VPD and a decreasing trend of SM (Fig. 7b). While TROPOMI SIF continued to increase as a result of increasing PAR, the grasslands in Region 2 already suffered functional depression due to thermal and/or water stress. Similarly, in Region 1 on October 24th, TROPOMI SIF also showed a slight increase due to an increase in PAR, while TROPOMI SIF yield showed a reduction related to increased Tair and VPD and decreased SM (Fig. 7a). This underscores the unique and valuable functional information contained in TROPOMI SIF for stress early detection and preparedness. In addition, during the second timestamp of all three selected windows (i.e., October 16th for Region 1 and 2, and November 25th for Region 3), when TROPOMI SIF had a strong increase, TROPOMI SIF yield also increased under favourable conditions (e.g., relatively lower Tair and VPD and higher SM). This might explain the stronger vegetation signals observed in TROPOMI SIF compared to other datasets (with less increase in SIF yield, Fig. S12) during the peak growing season (Fig. 4). The environmental effect on vegetation function is further demonstrated by the strong correlation between SIF yield derived from TROPOMI and meteorological variables, especially Tair and VPD (Fig. 14d – 14e). This highlights the unique capability of TROPOMI SIF for vegetation monitoring and ultimately carbon budget quantification.

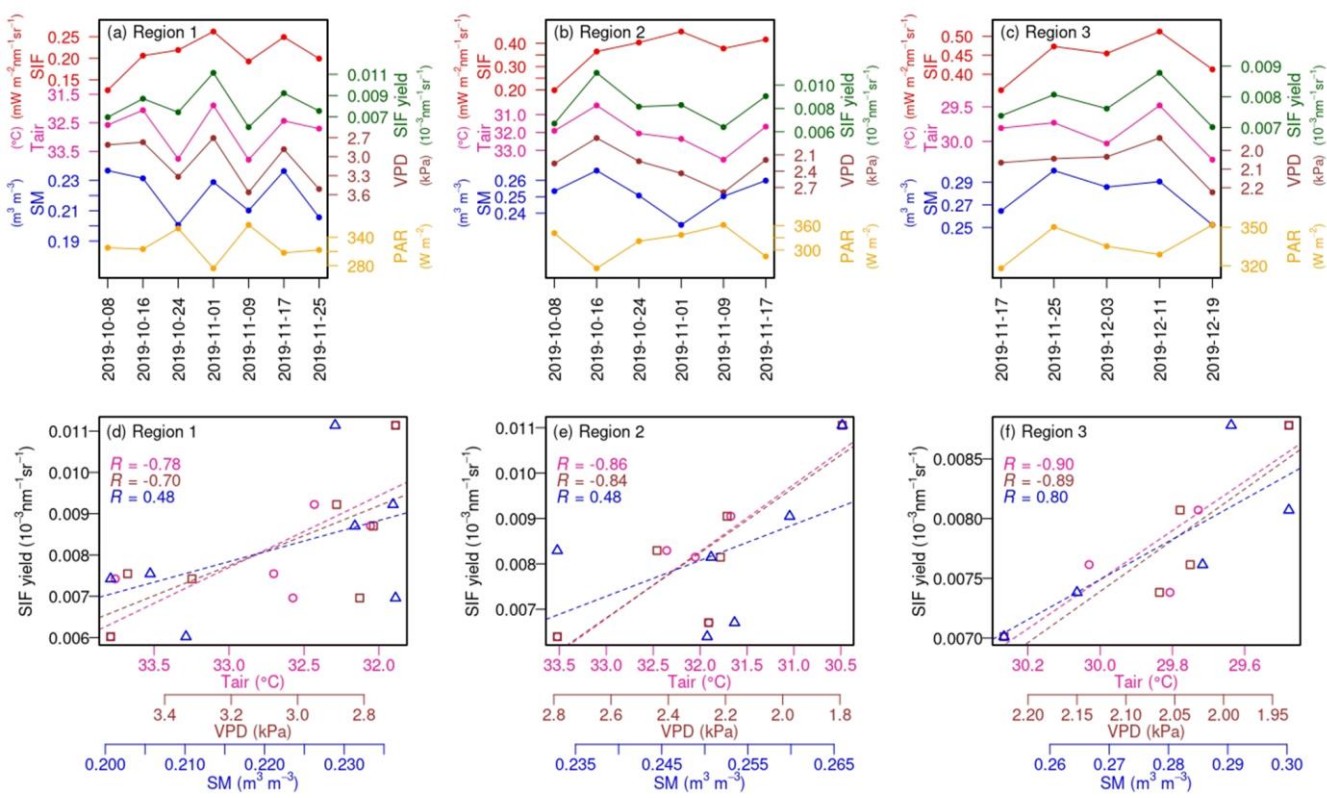

**Figure 7: (a) – (c)** Intra-seasonal variations of TROPOMI SIF, TROPOMI SIF yield, Tair, VPD, SM and PAR for the three sub-
domains during the selected time windows (Fig. 4). The y axes for Tair and VPD are reversed for visual clarity. The x axis labels
represent the starting date of each 8-day interval. **(d) – (f)** Scatterplots between TROPOMI SIF yield and Tair (pink circle), VPD
(brown square), and SM (blue triangle) for the three sub-domains during the selected time windows. The dashed lines represent
fitted linear regression lines, with correlation coefficients (*R*) noted in the upper left of each panel.

## 4 Discussion

### 4.1 Dryland intra-seasonal vegetation dynamics under excessive precipitation

Dryland ecosystems are characterized by highly variable vegetation dynamics in response to environmental drivers at short
time scales. Monitoring and understanding their behavior under different environmental conditions is critical to predict the
fate of the global terrestrial carbon sink as well as support the livelihood of billions of people who live therein. In this study,
we revealed the fast-changing intra-seasonal vegetation dynamics of HoA drylands under excessive precipitation, utilizing
several high-temporal-resolution SIF and VI datasets, especially TROPOMI SIF with unprecedented daily revisit frequency
for satellite SIF retrieval. As revealed by TROPIMI SIF, we found interesting temporal dynamics of dryland vegetation

under excessive precipitation at both site and regional levels. At the Kapiti site, there was not only a more pronounced vegetation signal (Fig. 2), but also complex phenological and physiological changes happening. In response to extreme soil moisture and rainfall amount and duration, two separate growth cycles occurred within a single rainy season, accompanied with a reduction in SIF (possibly also productivity) during the transition period between the two growth cycles (Fig. 3). Turner et al. (2019) also reported a double-peak SIF signal within one single growing season, due to different phenology of grasses and evergreen forests, while MODIS VIs failed to capture. At the regional scale, TROPOMI SIF showed highly dynamic week-to-week variations (Figs. 4-6) functionally up- and down- regulated by environment fluctuations (e.g., Tair, VPD and SM, Fig. 7) for all three selected sub-domains with distinct land cover types, precipitation variations and vegetation responses. Such short-term plant stresses and recoveries suggested strong environmental constraints (e.g., thermal and/or water stress) on dryland ecosystem functions, even during a rainy season with anomalously high precipitation. Such vegetation dynamics were not able to be unraveled by other datasets (e.g., MODIS NIRv). The findings in this study may alter our knowledge about monitoring of dryland ecosystems and their phenological and physiological responses to a changing climate under future projections, which inspires further investigation.

## 4.2 Advantages of SIF over VIs in revealing intra-seasonal dynamics

Satellite SIF has emerged as a promising proxy for inferring spatiotemporal dynamics of photosynthetic activities from canopy to global scales. Numerous studies have compared the capability of SIF and VIs in characterizing the temporal dynamics of GPP at seasonal or inter-annual scales. SIF has unique mechanistic advantages as it is emitted from the core of the photosynthetic machinery and therefore contains additional functional information beyond the structural information usually carried by VIs. On the other hand, SIF has its practical limitations (e.g., comparatively coarser spatial and temporal resolutions, and higher measurement/retrieval noise) relative to the greenness-based VIs that are much easier to retrieve. The general consensus from previous studies is that satellite SIF has overall similar performance to greenness-based VIs at seasonal cycles and beyond. This is especially the case for crops and deciduous forests where seasonal variations of structure (e.g., leaf area index and pigment content) are dominant (Dechant et al. 2020; Yang et al. 2015). In ecosystems where functional changes (e.g., leaf physiology) play a more impactful role compared to structural changes (e.g., evergreen conifers), more pronounced advantages have been found in SIF over greenness-based VIs in inferring GPP seasonal dynamics (Magney et al., 2019; Pierrat et al., 2022).

While previous studies have been mostly focused on evaluation at the seasonal scale and beyond, this study highlighted the differences in shorter time scales, e.g., intra-seasonal. To achieve this, we took advantage of TROPOMI SIF (with daily revisit frequency) and employed HoA drylands (with highly dynamic vegetation changes in response to the environment) as a testbed. We found that only TROPOMI SIF revealed fast-changing phenological and physiological variations at both site and regional levels, while MODIS NIRv failed to capture them, despite that the latter is provided at high temporal resolution. This is mainly because, at a temporal scale of several days to weeks especially during the peak growing season, the functional changes (as contained in SIF) in response to short-term environmental fluctuations are dominant compared to

structural changes (as represented by NIRv), as greenness remains relatively constant (Daumard et al., 2010; Martini et al., 2022). This is in analogy to the case of evergreen conifers at the seasonal scale (Magney et al., 2019; Pierrat et al., 2022). Such differences suggest that SIF contains unique mechanistic value in estimating carbon sequestration, monitoring vegetation status, detecting early plant stress, and understanding climate-vegetation interactions at short time scales.

A recent study by Wang et al. (2022c) evaluated the ability of satellite SIF and NIRv in capturing the seasonal variation of GPP in dryland ecosystems, and found that NIRv performed better than SIF for low-productivity sites, likely because of the low signal-to-noise ratio of SIF retrievals. This does not necessarily contradict the findings of our study. First, Wang et al. (2022c) examined the performance of SIF and NIRv at the seasonal scale over about two years, when SIF may have only marginal advantages in inferring function-related variations that are overwhelmed by structure-related variations. In contrast,
our study focused on intra-seasonal variations, when functional changes have a stronger impact. Second, our evaluation is conducted on a relatively wet period when vegetation signals are strong, therefore the data noise has less influence on the retrieved vegetation signals.

## 4.3 Deficiencies of reconstructed SIF products

The native satellite SIF retrievals have been long suffering coarse spatial and/or temporal resolutions, large data noise, and
short time spans. It has been hoped that the reconstructed SIF products that are derived from the native SIF retrievals could overcome these practical limitations, improving the capability of satellite SIF in depicting vegetation dynamics across scales. Indeed, there have been many efforts in the past years in developing such products (e.g., Duveiller and Cescatti 2016; Zhang et al., 2018; Li and Xiao, 2019; Yu et al., 2019; Wen et al., 2020; Ma et al., 2020, 2022; Chen et al., 2022; Wang et al., 2022b). However, this study found that these reconstructed SIF products (i.e., based on OCO-2 or TROPOMI) resembled the
spatiotemporal patterns of MODIS NIRv and were unable to characterize the complex fast-changing intra-seasonal dynamics (that were successfully captured by TROPOMI SIF), although these products were provided at fine temporal resolutions (e.g., 4 days for CSIF).

This may be explained by two aspects of generating these reconstructed SIF products. First, the native SIF retrievals used for SIF reconstruction must contain the signals of fast-changing intra-seasonal vegetation dynamics. However, the native SIF
retrievals from OCO-2 (with a 16-day revisit cycle) most likely miss these fast-changing signals, especially during the rainy seasons when clouds may exacerbate the issue. In contrast, the native SIF retrievals from TROPOMI (with daily revisit frequency) can track the complex intra-seasonal vegetation dynamics. This highlights the demand for native SIF retrievals with high temporal resolutions, e.g., several upcoming geostationary missions such as Tropospheric Emissions: Monitoring of Pollution (TEMPO) and the Copernicus Sentinel-4, which may greatly facilitate capturing vegetation dynamics at fine
temporal scales and understanding climate-vegetation interactions.

Second, the SIF reconstruction must faithfully preserve the spatiotemporal variations of native SIF. The procedure of SIF reconstruction is essentially a mapping from the ancillary datasets to SIF with calibrated relationships (Sect. 2.3). Most of the SIF reconstruction studies calibrated the relationships based on evaluation across all timestamps and all pixels,

when/where the structural changes overwhelmingly dominate the variations, therefore whether the important functional

information is preserved is not effectively evaluated. For example, the SIF yield calculated from RTSIF and GOSIF is largely dampened compared to that from TROPOMI SIF, which leads to flatter intra-seasonal variations in RTSIF and GOSIF, therefore a much weaker environmental sensitivities (Fig. S12), and lower consistency with in situ SIF (Fig. 2). This is however not contradicted with the high consistency between RTSIF and TROPOMI (e.g., $R^2$ = 0.907, regression slope = 1.001, reported in Chen et al., 2022), probably as a result of both correlating with absorbed PAR. To preserve the functional

information of the native SIF retrievals in the reconstructed SIF, one general idea is to impose a stronger constraint from the native SIF during the SIF reconstruction. For example, Wen et al. (2020) demonstrated that by stratifying the models in time and space, the reconstructed SIF could be better constrained by spatiotemporal variations of the native SIF and therefore be capable of capturing the functional changes. Another possible approach is to calculate the differences between the reconstructed SIF and the native SIF and redistribute the prediction residuals to the reconstructed SIF. Recently, Ma et al.

(2022) utilized such an approach to reconstruct high-resolution SIF from the Global Ozone Monitoring Experiment-2 (GOME-2). With the redistribution of prediction residuals, the reconstructed SIF showed a greater consistency with the native GOME-2 SIF. However, such approaches can only be applied to timestamps/regions when/where the native SIF retrievals are available. It could be challenging to make such adjustments for the extrapolated SIF when/where the native SIF retrievals do not exist (e.g., TROPOMI before 2018, spatial gaps for OCO-2).

**4.4 Limitations and future work**

Nonetheless, there are still several limitations in this study, which warrants future work. First, while this study utilized the HoA dryland ecosystems as a testbed to evaluate the capability of different satellite SIF and VI products in capturing intra-seasonal dynamics, such comparison could be further conducted for other dryland regions or other vegetation types towards a more comprehensive evaluation. Second, limited by the scarcity of in-situ data, the intra-seasonal variations of SIF inferred

in this study were not directly linked to ecosystem productivity. Such evaluation could be conducted in regions with more in-situ data, e.g., flux tower measurements, as complementary assessment. Third, while this study evaluated the intra-seasonal variations inferred from different products in a qualitative way, further quantitative analysis can be done in the future work, e.g., to quantify the climate sensitivities of vegetation carbon dynamics.

**5 Conclusions**

Accurately monitoring the fast-changing vegetation dynamics of dryland ecosystems has been critical for understanding their climate sensitivities and informing climate risk management. In this study, we evaluated the advantages of SIF over greenness-based VIs in characterizing intra-seasonal (i.e., from days to weeks) vegetation dynamics, utilizing dryland ecosystems (e.g., shrublands and grasslands) in the Horn of Africa (HoA) as a testbed. At both site and regional levels, we found that TROPOMI SIF revealed fast-changing phenological and physiological variations at the intra-seasonal scale, while

MODIS NIRv and several reconstructed SIF products did not. Specifically, at the site level, our results showed that TROPOMI SIF revealed two separate within-season growth cycles in response to extreme soil moisture and rainfall amount and duration, which was corroborated by in situ SIF measurements and Phenocam images. At the regional level, TROPOMI SIF and SIF yield exhibited highly dynamic week-to-week variations in both shrublands and grasslands, driven by environmental fluctuations (e.g., Tair, VPD, SM). MODIS NIRv could not capture such fast-changing intra-seasonal variations but remained relatively stable during the same period. Interestingly, the machine-learning reconstructed SIF products were unable to characterize such intra-seasonal dynamics either, despite their approximately weekly temporal resolutions, rooted in insufficient temporal granularity of their original SIF retrievals and inadequate constraints from native SIF retrievals during the reconstruction. Our results indicate that SIF carries mechanistic advantages over NIRv in monitoring fast-changing intra-seasonal dynamics for dryland ecosystems, but high-temporal resolution SIF is essential to capture such complicated patterns. This study generates novel and important insights for developing effective real-time vegetation monitoring systems to understand carbon dynamics and inform climate risk management.

**Data and Code Availability**

In situ SIF is available upon request to micol.rossini@unimib.it. TROPOMI_ESA can be accessed from http://ftp.sron.nl/open-access-data-2/TROPOMI/tropomi/sif. TROPOMI_Caltch can be accessed from ftp://fluo.gps.caltech.edu. RTSIF can be accessed from https://doi.org/10.6084/m9.figshare.19336346.v2. CSIF can be accessed from https://osf.io/8xqy6. GOSIF can be accessed from http://data.globalecology.unh.edu/data/GOSIF_v2. SIF_oco2_005 can be accessed from https://daac.ornl.gov/cgi-bin/dsviewer.pl?ds_id=1863. MODIS MCD43A4 can be accessed from https://lpdaac.usgs.gov/products/mcd43a4v061/. CHIRPS precipitation data can be downloaded from https://data.chc.ucsb.edu/products/CHIRPS-2.0/. ESA-CCI soil moisture data can be accessed from https://www.esa-soilmoisture-cci.org/data. MERRA-2 reanalysis can be accessed from https://disc.gsfc.nasa.gov/datasets/. MODIS LC can be accessed from https://lpdaac.usgs.gov/products/mcd12c1v061. Code for analyses and figures are available at: https://github.com/JiamingWen/Kapiti_intraseasonal.

**Author contribution**

JW and YS contributed to the conceptualization of this paper. JW and GT curated data, did formal analysis and generated all figures. MR, CP, LM, SL and FPF curated the in-situ data and maintained the research station. YS, MR and FPF are responsible for supervision and resources. JW and GT prepared the original draft. All authors reviewed and edited the manuscript.

**Competing interests**

Lutz Merbold is a member of the editorial board of journal Biogeosciences.

**Acknowledgments**

We thank Ilona Gluecks (ILRI), ILRI Mazingira team and Kapiti farm staff for their support in the setup and maintenance of the experimental site. We thank Tommaso Julitta, Andreas Burkart and Paul Näthe (JB Hyperspectral Devices) for their remote support in setting up and maintaining the FloX system. JW and YS acknowledge support from NASA-CMS (80NSSC21K1058), NASA-FINESST (80NSSC20K1646). JW also acknowledges support from NASA-ABoVE 550 (80NSSC22K1253). LM and SL acknowledge support from the German Federal Ministry for Economic Cooperation and Development (BMZ issued through GIZ) through the "Programme of Climate Smart Livestock" (PCSL, Programme No. 2017.0119.2) as well as funding from the EU DeSIRA project "Earth observation and environmental sensing for climate-smart sustainable agropastoral ecosystem transformation in East Africa" (ESSA). LM further acknowledges funding received from the European Union's Horizon Europe Programme (grant agreement number 101058525) for the project "Knowledge 555 and climate services from an African observation and Data research Infrastructure (KADI)". SL would like to thank all funders who supported this research through their contributions to the CGIAR Trust Fund and the OneCGIAR research initiatives on Livestock and Climate and MITIGATE+. JW would like to acknowledge the helpful discussion with Dr. Joe Berry, Dr. Jeff Dukes and Andrea Nebhut at Carnegie Institution for Science.

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
