# Peer review of "Detection of Fast-Changing Intra-seasonal Vegetation Dynamics of Drylands Using Solar-Induced Chlorophyll Fluorescence (SIF)"

_EGUsphere, 2024_

## Author Response (AR1)

We sincerely thank the reviewers for their valuable comments and suggestions. Please find our responses below in blue. The revisions of the manuscript are quoted in *red italics*, with their corresponding sections and line numbers in the revised manuscript marked in **bold red**.

**Reviewer 1**

[RC1-1] The paper by Wen et al. examines the performance of in-situ and satellite-derived solar-induced fluorescence (SIF) and vegetation indices in tracking intra-seasonal vegetation dynamics in an African dryland ecosystem. Overall, the paper is well-written and conveys an important finding: TROPOMI SIF aligns well with in-situ SIF measurements in the Horn of Africa (HoA), while other reflectance-based vegetation indices may overlook certain subseasonal vegetation dynamics. These findings have potential implications for understanding dryland carbon fluxes. Given the good shape of the manuscript, I have a few comments to further improve the paper.

[AR1-1] We thank the reviewer for appreciating our work and providing constructive comments that helped to improve this manuscript. Please find below our point-by-point response and revisions.

Introduction:

[RC1-2] The authors provide an insightful introduction to the importance of drylands and the role of SIF and vegetation indices. However, there is limited discussion on the existing literature concerning SIF applications for tracking dryland GPP or drought/heat stresses. I suggest the authors expand this to provide a more comprehensive background on SIF use in dryland ecosystems.

[AR1-2] We appreciate this suggestion. While we have described SIF's mechanistic advantages over VIs in general in **Line 69-71**, we have added more specific descriptions on the applications of SIF in monitoring for dryland ecosystem dynamics, as this reviewer suggested. Please find our revised descriptions in **Line 71-77**: "*In addition, since SIF signal comes only from active vegetation, it is less susceptible to the brightness of soil background, unlike reflectance-based VIs (Huete et al., 2002). These characteristics make SIF a unique observational signal for inferring photosynthetic dynamics for dryland ecosystems. For example, SIF has demonstrated a superior capability in accurately depicting dryland ecosystem phenology (Wang et al., 2019) as well as capturing seasonal variations (Wang et al., 2022c) and interannual variations (Smith et al., 2018) of in situ gross primary production (GPP). Furthermore, it has facilitated many applications in drought detection and ecosystem restoration in drylands (Robinson et al., 2019; Mengistu et al. 2021, Constenla-Villoslada et al., 2022).*"

*Huete, A., Didan, K., Miura, T., Rodriguez, E. P., Gao, X., and Ferreira, L. G.: Overview of the radiometric and biophysical performance of the MODIS vegetation indices, Remote Sens Environ, 83, 195–213, https://doi.org/10.1016/S0034-4257(02)00096-2, 2002.*

*Wang, C., Beringer, J., Hutley, L. B., Cleverly, J., Li, J., Liu, Q., and Sun, Y.: Phenology Dynamics of Dryland Ecosystems Along the North Australian Tropical Transect Revealed by*

*Satellite Solar-Induced Chlorophyll Fluorescence, Geophys Res Lett, 46, 5294–5302, https://doi.org/10.1029/2019GL082716, 2019.*

*Wang, X., Biederman, J. A., Knowles, J. F., Scott, R. L., Turner, A. J., Dannenberg, M. P., Köhler, P., Frankenberg, C., Litvak, M. E., Flerchinger, G. N., Law, B. E., Kwon, H., Reed, S. C., Parton, W. J., Barron-Gafford, G. A., and Smith, W. K.: Satellite solar-induced chlorophyll fluorescence and near-infrared reflectance capture complementary aspects of dryland vegetation productivity dynamics, Remote Sens Environ, 270, https://doi.org/10.1016/j.rse.2021.112858, 2022c.*

[RC1-3] Line 78: The IPCC reference can be more specific

[AR1-3] We appreciate this suggestion and have specified details for this reference. We have changed it to *"IPCC Working Group II, Chapter 9, 2022"* in **Line 85**.

Method:

[RC1-4] When filtering satellite SIF measurements, it is unclear whether the authors account for the uncertainties in SIF retrievals (e.g., standard deviations of SIF) provided by the products.

[AR1-4] Thank you for this constructive comment. We did not account for the uncertainties of SIF retrievals in our original submission, as previous studies have rarely used the SIF retrieval uncertainty for quality control. In the revision, we tested the robustness/sensitivity of our results to SIF retrieval uncertainties. While the retrieval error for TROPOMI_ESA (743–758 nm fitting window) is typically 0.5 mW m$^{-2}$ sr$^{-1}$ nm$^{-1}$(Guanter et al., 2021), we tested with a stricter threshold (0.4 mW m$^{-2}$ sr$^{-1}$ nm$^{-1}$) and found that the temporal pattern remains largely unchanged, as shown in the figure below.

[Figure]

We have added this panel to **Fig S3** along with descriptions to the corresponding text in the main manuscript (**Line 296-298**).

*"This double-peak pattern in TROPOMI SIF held, regardless of sources of TROPOMI data, fitting windows used for SIF retrievals, or quality filtering criteria (e.g., SZA, CF, and retrieval error) (Fig. S3a-S3d)."*

Results:

[RC1-5] There are negative SIF values in both the in-situ observations and the TROPOMI SIF retrievals, as shown in Fig. 2. Since SIF should theoretically be positive, the authors should explain the source of these negative values, such as measurement errors or retrieval limitations.

[AR1-5] This is a great point, but we have to argue that negative SIF values should be retained in the analysis, a practice that is commonly used in the SIF community (e.g., Sun et al., 2018; Doughty et al., 2022) and suggested for TROPOMI SIF applications (Guanter et al., 2021). This is because negative SIF values are mainly a consequence of measurement and retrieval noises, even though theoretically SIF should be positive. If removing these negative values during spatial/temporal aggregation, one would introduce artificial positive biases. In principle, if the noises are random, the average of SIF value in a non-vegetated area should be zero, i.e., average of both positive and negative values.

Doughty, R., Kurosu, T. P., Parazoo, N., Köhler, P., Wang, Y., Sun, Y., and Frankenberg, C.: Global GOSAT, OCO-2, and OCO-3 solar-induced chlorophyll fluorescence datasets, Earth Syst. Sci. Data, 14, 1513–1529, https://doi.org/10.5194/essd-14-1513-2022, 2022.

Sun, Y., Frankenberg, C., Jung, M., Joiner, J., Guanter, L., Köhler, P., and Magney, T.: Overview of Solar-Induced chlorophyll Fluorescence (SIF) from the Orbiting Carbon Observatory-2: Retrieval, cross-mission comparison, and global monitoring for GPP, Remote Sens Environ, 209, 808–823, https://doi.org/10.1016/j.rse.2018.02.016, 2018.

To clarify this point, we have added descriptions in **Line 265-270**: *"Note that some negative values appear in FloX SIF and TROPOMI SIF especially during the time periods with weak SIF signals, as a result of measurement and retrieval noise (Guanter et al., 2021). These negative values are retained in the evaluation to avoid an artificial positive bias in spatial and temporal aggregation."*

[RC1-6] Fig. 1(h): The legend with dotted lines does not match the lines in the figure.

[AR1-6] Thanks for pointing this out. We have revised the legend to match the lines in the figure.

[Figure]

*Figure 1h.* Time series of precipitation at Kapiti during 2019-2020 (dashed) and 2020-2021 (dotted), when in situ SIF was collected, compared to the multi-year mean (2011-2020, solid). The shade denotes one standard deviation of monthly precipitation during 2011 and 2020. The lengths of SR and LR seasons are marked on the x-axis in light blue.

[RC1-7] Fig. 7: It might be useful to include a scatter plot showing SIF yield against variables like VPD, soil moisture, or Tair to illustrate how much environmental factors drive variation in SIF yield.

[AR1-7] Thanks for this constructive feedback. We have added scatterplots between TROPOMI SIF yield and environmental factors in **Fig. 7 (d) - (f)**.

[Figure]

Figure 7: (a) – (c) Intra-seasonal variations of TROPOMI SIF, TROPOMI SIF yield, Tair, VPD, SM and PAR for the three sub-domains during the selected time windows (Fig. 4). The y axes for Tair and VPD are reversed for visual clarity. The x axis labels represent the starting date of each 8-day interval. (d) – (f) Scatterplots between TROPOMI SIF yield and Tair (pink circle), VPD (brown square), and SM (blue triangle) for the three sub-domains during the selected time windows. The dashed lines represent fitted linear regression lines, with correlation coefficients (*R*) noted in the upper left of each panel.

We have also added text descriptions in **Line 399-401**: *"The environmental effect on vegetation function is further demonstrated by the strong correlation between SIF yield derived from TROPOMI and meteorological variables, especially Tair and VPD (Fig. 14d – 14e)."*

Discussion:

[RC1-8] The authors could further discuss the performance of SIF in tracking dryland vegetation dynamics as reported in previous studies. For instance, Wang et al. (2022) found that NIRv performed better than SIF in capturing GPP in western U.S. drylands due to noise in SIF signals. The results of this study appear to differ somewhat from those of previous studies.

Wang, X., Biederman, J.A., Knowles, J.F., Scott, R.L., Turner, A.J., Dannenberg, M.P., Köhler, P., Frankenberg, C., Litvak, M.E., Flerchinger, G.N. and Law, B.E., 2022. Satellite solar-induced chlorophyll fluorescence and near-infrared reflectance capture complementary aspects of dryland vegetation productivity dynamics. *Remote sensing of environment*, *270*, p.112858.

[AR1-8] This is a great point. Indeed, we fully agree that the mechanistic advantages of SIF over VIs may be offset by its large measurement/retrieval noise, as we discussed in **Section 4.2 Line 436-437**, *"On the other hand, SIF has its practical limitations (e.g., comparatively coarser spatial and temporal resolutions, and higher measurement/retrieval noise) relative to the greenness-based VIs that are much easier to retrieve."*

In addition, we have explicitly compared and discussed the different conclusions drawn from their and our studies. We have added a new paragraph in **Section 4.2 Line 455-462**: *"A recent study by Wang et al. (2022c) evaluated the ability of satellite SIF and NIRv in capturing the seasonal variation of GPP in dryland ecosystems, and found that NIRv performed better than SIF for low-productivity sites, likely because of the low signal-to-noise ratio of SIF retrievals. This does not necessarily contradict the findings of our study. First, Wang et al. (2022c) examined the performance of SIF and NIRv at the seasonal scale over about two years, when SIF may have only marginal advantages in inferring function-related variations that are overwhelmed by structure-related variations. In contrast, our study focused on intra-seasonal variations, when functional changes have a stronger impact. Second, our evaluation is conducted on a relatively wet period when vegetation signals are strong, therefore the data noise has less influence on the retrieved vegetation signals."*

**Reviewer 2**

[RC2-1] This manuscript examines the potential of Solar-Induced Fluorescence (SIF) in detecting fast changes in vegetation function and structure in dryland ecosystems. The authors use the Horn of Africa as an example to compare the performance of different satellite SIF data products in representing the fasting-changing intra-seasonal vegetation dynamics in an abnormally wet year. First, the authors cross-compared the satellite SIF products against tower-based SIF. They found that native satellite SIF data matches tower-based SIF best. The reconstructed SIF products do not show intra-seasonal changes observed from the tower. Then, the authors use climate data and field images to explain the observed inter-seasonal changes. The results show that the native satellite SIF data is mechanistically linked to the vegetation function, which makes this data a good candidate for future real-monitoring vegetation dynamics. The manuscript is overall well written. Here are some major comments I have:

[AR2-1] We thank the reviewer for appreciating our work and providing constructive comments that helped to improve this manuscript. Please find below our point-by-point response and revisions.

[RC2-2] I think the manuscript is missing some detailed explanation of the choice of temporal scaling, especially since the temporal resolution is a key component in this manuscript. All results are presented as 8-day averages. However, the manuscript does not explain why 8-day is picked given other options are available. This may not be intuitive for readers who are not familiar with TROPOMI.

[AR2-2] Thank you for this valuable suggestion. We fully agree that temporal resolution is a key component in this work. In this work, we selected the 8-day resolution for the intra-seasonal analysis at both site and regional scales (Sect. 3.2 and 3.3). The 8-day resolution balances the tradeoff between reducing the measurement noise of TROPOMI SIF and in situ SIF and still preserving the fast-changing intra-seasonal dynamics, the major focus of this study. In addition, the 8-day resolution also matches the temporal resolution of the reconstructed SIF (GOSIF and RTSIF).

To address this comment, we have added a dedicated section **Sect. 2.5 Spatial and temporal matching criteria** to justify our rationale and to detail the approaches we used for spatial and temporal matching (this information was in Supplementary Materials in the original submission).

*2.5 Spatial and temporal matching criteria*

*We employed multiple spatial and temporal matching criteria for the intercomparison among different SIF datasets (Sect. 3.1) and for the analysis of intra-seasonal vegetation dynamics (Sect. 3.2 and 3.3). The principle is as follows: for the SIF intercomparison against in situ SIF, we aimed to ensure the best spatial/temporal consistency between in situ SIF and each satellite SIF dataset to be evaluated; for the intra-seasonal analysis, we attempted to ensure the*

*spatial/temporal consistency among all the datasets (including SIF and other ancillary variables) so that all the variables refer to the same spatial domains and time intervals.*

**2.5.1 Spatial and temporal matching criteria for SIF intercomparison**

***Spatial matching:*** *for comparison with in situ SIF measurements, TROPOMI was regridded to 0.15° pixel (as explained in Sect. 2.3) centered at the tower location. For the reconstructed SIF products, we extracted the value of the 0.05° pixel where the tower is located to minimize the difference in spatial scales.*

***Temporal matching:*** *For the paired comparison between in situ SIF and TROPOMI, we selected the quality-filtered in situ SIF observations (Sect. 2.2) collected within a time window of ±30 minutes with respect to the overpass time of each TROPOMI observation. The selected measurements were averaged after applying the daily-correction factor based on the SZA, which was also applied to the TROPOMI SIF. The TROPOMI observations for which no in situ SIF observations were available in the ±30 minutes time window were discarded. In total, 64 data pairs were used for comparison.*

*For the paired comparison between in situ SIF and the reconstructed SIF products, we extracted the quality-filtered in-situ SIF within a ±30 minutes time window centered at the OCO-2 and TROPOMI nominal overpass time at the equator (i.e., 13:30 local solar time), applied the daily correction factor and then averaged the measurements across 4-, 8- or 16-day periods to match with the temporal resolution of the reconstructed products. In total, 67, 84, 40 and 55 data pairs were used for comparison for CSIF, GOSIF, SIF_oco2_005 and RTSIF, respectively.*

**2.5.2 Spatial and temporal matching criteria for intra-seasonal analysis**

*To ensure consistency among all datasets, we aggregated or resampled all datasets (e.g., in-situ SIF, TROPOMI SIF, MODIS NIRv, and climate variables) to the same 0.15° and 8-day resolutions. The 0.15° pixels were set so that the boundary of the 0.15° pixel around the tower was aligned with the 3 × 3 0.05° pixels of GOSIF and RTSIF closest to the tower. Therefore, this is slightly different from the 0.15° pixel for TROPOMI SIF described in Sect. 2.5.1. The 8-day resolution was selected, (1) to reduce the measurement noise of TROPOMI SIF and in situ SIF while preserving the fine-scale intra-seasonal temporal variations, and (2) to match the coarser temporal resolution of GOSIF and RTSIF. For the analysis of intra-seasonal dynamics at Kapiti (Sect. 3.1), we selected the quality-filtered in situ SIF observations (Sect. 2.2) collected within a time window of ±30 minutes with respect to the overpass time of TROPOMI and applied a daily-correction factor based on the SZA to convert them into daily values. The daily values were then aggregated to the same 8-day intervals.*

The descriptions about the selection of 8-day resolution are quoted here:

*"The 8-day resolution was selected, (1) to reduce the measurement noise of TROPOMI SIF and in situ SIF while preserving the fine-scale intra-seasonal temporal variations, and (2) to match the coarser temporal resolution of GOSIF and RTSIF."* **(Line 244-246)**

[RC2-3] The manuscript did not present how SIF from different spatial resolutions and footprints are cross-compared, nearest pixel?

[AR2-3] Please see our response above. The spatial matching criteria were described in Supplementary Materials in the original submission and now have been moved to a dedicated section (**Sect. 2.5**) in the main manuscript. We employed different strategies of spatial matching for the analysis of SIF intercomparison (Sect. 3.1) and the analysis of intra-seasonal vegetation dynamics (Sect. 3.2 and 3.3). Our principle is: *"for the SIF intercomparison against in situ SIF, we aimed to ensure the best spatial/temporal consistency between in situ SIF and each satellite SIF dataset to be evaluated; for the intra-seasonal analysis, we attempted to ensure the spatial/temporal consistency among all the datasets (including SIF and other ancillary variables) so that all the variables refer to the same spatial domains and time intervals."* (**Line 221-225**)

Specifically, for SIF intercomparison, *"for comparison with in situ SIF measurements, TROPOMI was regridded to 0.15° pixel (as explained in Sect. 2.3) centered at the tower location. For the reconstructed SIF products, we extracted the value of the 0.05° pixel where the tower is located to minimize the difference in spatial scales."* (**Line 227-229**) And in Sect. 2.3, we described our rationale of re-gridding TROPOMI SIF to 0.15°: *"0.15° was selected to include enough soundings for spatial aggregation to reduce measurement noise while maintaining overall representativeness of the area around the tower (Fig. S1)."* (**Line 179-181**)

For intra-seasonal analysis, *"To ensure consistency among all datasets, we aggregated or resampled all datasets (e.g., in-situ SIF, TROPOMI SIF, MODIS NIRv, and climate variables) to the same 0.15° and 8-day resolutions. The 0.15° pixels were set so that the boundary of the 0.15° pixel around the tower was aligned with the 3 × 3 0.05° pixels of GOSIF and RTSIF closest to the tower. Therefore, this is slightly different from the 0.15° pixel for TROPOMI SIF described in Sect. 2.5.1."* (**Line 241-244**)

[RC2-4] I think authors should cite papers to support their explanations for physiological changes, e.g., lines 270-271. In some locations, it would be better to include drylands/Africa-related references in addition to global-scale references, e.g., line 50.

[AR2-4] Thanks for these great suggestions.

For references to support our explanations for physiological changes in the result section, we have revised our descriptions and added relevant references in **Line 315-318**: *"The grass progressed to the reproductive stage during early and mid December (Fig., S5b, Cheng et al.,*

*2020; Zhang et al., 2023), resulting in a gradual decrease in the photosynthetic activity, possibly because of nutrient remobilization and carbohydrate sink limitation (Tejera-Nieves et al., 2023)."*

*Cheng, Y., Vrieling, A., Fava, F., Meroni, M., Marshall, M., and Gachoki, S.: Phenology of short vegetation cycles in a Kenyan rangeland from PlanetScope and Sentinel-2, Remote Sens Environ, 248, 112004, https://doi.org/10.1016/j.rse.2020.112004, 2020.*

*Zhang, Z., Zhang, Z., Hautier, Y., Qing, H., Yang, J., Bao, T., Hajek, O. L., and Knapp, A. K.: Effects of intra-annual precipitation patterns on grassland productivity moderated by the dominant species phenology, Front Plant Sci, 14, 1142786, https://doi.org/10.3389/FPLS.2023.1142786/BIBTEX, 2023.*

*Tejera-Nieves, M., Abraha, M., Chen, J., Hamilton, S. K., Robertson, G. P., and Walker James, B.: Seasonal decline in leaf photosynthesis in perennial switchgrass explained by sink limitations and water deficit, Front Plant Sci, 13, 1023571, https://doi.org/10.3389/FPLS.2022.1023571/BIBTEX, 2023.*

For references related to Africa or drylands in the introduction, we have to argue that, in the original submission, we had several references related to:

- east Africa (e.g., Williams et al., 2012; Lyon and Dewitt, 2012; Funk et al., 2015; Ngoma et al., 2021; Matanó et al., 2022; Pricope et al. 2013; Beal et al. 2023);
- drylands (e.g., Prăvălie 2016; Poulter et al. 2014; Ahlström et al. 2015; Piao et al. 2020; Yao et al. 2020; Lian et al., 2021; Huang et al., 2015, Huang et al. 2017; Smith et al., 2019; Zhang et al. 2020a, 2022; Wang et al., 2022a; Adams et al., 2021; Smith et al., 2018; Robinson et al., 2019; Mengistu et al. 2021, Constenla-Villoslada et al., 2022).

Some of them are specifically focused on African drylands (e.g., Robinson et al., 2019; Mengistu et al. 2021, Constenla-Villoslada et al., 2022).

In the revision, we have included more references about SIF applications in inferring dryland vegetation dynamics in **Line 73-76**: *"For example, SIF has demonstrated a superior capability in accurately depicting dryland ecosystem phenology (Wang et al., 2019) as well as capturing seasonal variations (Wang et al., 2022c) and interannual variations (Smith et al., 2018) of in situ gross primary production (GPP)."*

*Wang, C., Beringer, J., Hutley, L. B., Cleverly, J., Li, J., Liu, Q., and Sun, Y.: Phenology Dynamics of Dryland Ecosystems Along the North Australian Tropical Transect Revealed by Satellite Solar-Induced Chlorophyll Fluorescence, Geophys Res Lett, 46, 5294–5302, https://doi.org/10.1029/2019GL082716, 2019.*

*Wang, X., Biederman, J. A., Knowles, J. F., Scott, R. L., Turner, A. J., Dannenberg, M. P., Köhler, P., Frankenberg, C., Litvak, M. E., Flerchinger, G. N., Law, B. E., Kwon, H., Reed, S. C., Parton, W. J., Barron-Gafford, G. A., and Smith, W. K.: Satellite solar-induced chlorophyll fluorescence and near-infrared reflectance capture complementary aspects of dryland*

*vegetation productivity dynamics, Remote Sens Environ, 270,*
*https://doi.org/10.1016/j.rse.2021.112858, 2022c.*

[RC2-5] The definition of vegetation function can be ambiguous. In line 57, Li et al., 2024 used "vegetation function" to include both physiology and structure. In this manuscript, vegetation function seems to only refer to physiology.

[AR2-5] Thanks for pointing this out. Indeed, in this manuscript, vegetation function refers to plant physiology (e.g., photosystem redox states, nonphotochemical quenching, electron transport rate, etc., all of which affect the efficiency of light use) and does not include leaf/canopy structure (e.g., leaf area, leaf angle, or pigment content, all of which affect light absorption and scattering). This terminology follows our previous review papers for SIF (Sun et al., 2023a, 2023b) along with other publications (e.g., Baldocchi et al., 2020; Dechant et al., 2020).

Baldocchi, D. D., Ryu, Y., Dechant, B., Eichelmann, E., Hemes, K., Ma, S., Sanchez, C. R., Shortt, R., Szutu, D., Valach, A., Verfaillie, J., Badgley, G., Zeng, Y., and Berry, J. A.: Outgoing Near-Infrared Radiation From Vegetation Scales With Canopy Photosynthesis Across a Spectrum of Function, Structure, Physiological Capacity, and Weather, J Geophys Res Biogeosci, 125, e2019JG005534, https://doi.org/10.1029/2019JG005534, 2020.

We have clarified this in **Line 55-59**: *"First, the former characterizes variations that are mainly driven by changes in vegetation function (i.e., leaf physiology, such as photosystem redox states, nonphotochemical quenching, electron transport rate, etc., all of which affect the efficiency of light use) (Gu et al., 2019; Han et al., 2022; Sun et al., 2023a), while the latter characterizes variations that are largely driven by changes in vegetation structure (e.g., leaf area, leaf angle, or pigment content, all of which affect light absorption and scattering) (Li et al., 2024)."*

[RC2-6] The manuscript introduces new terminology in results and discussion, such as NIRvP and SIFyield. They need to be better introduced before the results.

[AR2-6] Thanks for this great comment. In the revision, we have introduced NIRvP in the introduction by adding *"For example, NIRvP, the product of NIRv and photosynthetically active radiation (PAR), was found to be a robust structural proxy for photosynthesis (Dechant et al., 2022)."* in **Line 65-67**.

We have introduced SIF yield in **Sect. 2.3 Line 197-200**:

*"**SIF yield:** SIF yield carries information on plant physiological/functional variations in response to environmental changes (Sun et al., 2015; Yoshida et al. 2015; Yang et al., 2015; Miao et al., 2018; Magney et al., 2019; Sun et al., 2023a). In this study, to tease out the plant functional*

*variations from structural variations contained in the remotely sensed SIF signal, we derived SIF yield = SIF / PAR / NIRv, following Dechant et al., 2020."*

*Yang, X., Tang, J., Mustard, J. F., Lee, J.-E., Rossini, M., Joiner, J., Munger, J. W., Kornfeld, A., and Richardson, A. D.: Solar-induced chlorophyll fluorescence that correlates with canopy photosynthesis on diurnal and seasonal scales in a temperate deciduous forest, Geophys Res Lett, 42, 2977–2987, https://doi.org/10.1002/2015GL063201, 2015.*

*Miao, G., Guan, K., Yang, X., Bernacchi, C. J., Berry, J. A., DeLucia, E. H., Wu, J., Moore, C. E., Meacham, K., and Cai, Y.: Sun-Induced Chlorophyll Fluorescence, Photosynthesis, and Light Use Efficiency of a Soybean Field from Seasonally Continuous Measurements, J Geophys Res Biogeosci, 123, 610–623, 2018.*

*Magney, T. S., Bowling, D. R., Logan, B. A., Grossmann, K., Stutz, J., Blanken, P. D., Burns, S. P., Cheng, R., Garcia, M. A., Köhler, P., Lopez, S., Parazoo, N. C., Raczka, B., Schimel, D., and Frankenberg, C.: Mechanistic evidence for tracking the seasonality of photosynthesis with solar-induced fluorescence, Proceedings of the National Academy of Sciences, https://doi.org/10.1073/pnas.1900278116, 2019.*

*Sun, Y., Fu, R., Dickinson, R., Joiner, J., Frankenberg, C., Gu, L., Xia, Y., and Fernando, N.: Drought onset mechanisms revealed by satellite solar-induced chlorophyll fluorescence: Insights from two contrasting extreme events, J Geophys Res Biogeosci, 120, 2427–2440, https://doi.org/10.1002/2015JG003150, 2015.*

*Yoshida, Y., Joiner, J., Tucker, C., Berry, J., Lee, J. E., Walker, G., Reichle, R., Koster, R., Lyapustin, A., and Wang, Y.: The 2010 Russian drought impact on satellite measurements of solar-induced chlorophyll fluorescence: Insights from modeling and comparisons with parameters derived from satellite reflectances, Remote Sens Environ, 166, 163–177, https://doi.org/10.1016/j.rse.2015.06.008, 2015.*

Some typographical suggestions:

[RC2-7] Lines 215-216 are not clear and seem to miss a word.

[AR2-7] We have rephrased the sentence to improve its clarity: *"In situ SIF showed strong inter-annual variability, with a much stronger signal in the first year compared to the second year, driven by the difference in precipitation between the two years (Fig. 1h). It also exhibited pronounced intra-annual variations such as growth peaks during SR seasons (e.g., November 2019 - January 2020, December 2020), LR seasons (e.g., May 2020, June 2021), and a dry season with intermittent precipitation (February - March 2021, Fig. 1h) (Fig. 2a)."* in **Line 223-227**.

[RC2-8] Line 221 "...variation in SIF…", do you mean reconstructed SIF?

[AR2-8] Thanks for pointing this out. Yes, we were referring to the reconstructed SIF. We have clarified this in **Line 262-264** *"The reconstructed SIF products showed less frequent intra-seasonal variations, and their magnitudes of variations are sometimes inaccurate (e.g., the drop in December 2019 and the peak in February - March 2021), leading to lower correlation against FloX SIF$_{iFLD}$ compared to TROPOMI (r = 0.58-0.62, Fig. 2f-2i)."*

[RC2-9] Line 243, this line needs elaborations on why anomalous vegetation dynamics are challenging to measure. Is the anomaly or vegetation dynamics in general making it challenging?

[AR2-9] Thanks for this great comment. We have revised the sentence to avoid confusion: *"This period was chosen because excessive precipitation occurred during this SR season (i.e., 799 mm relative to the 2011-2020 average 343 ± 170 mm, Fig. 1h), leading to complex vegetation dynamics that can be challenging to be accurately characterized by satellite measurements. These challenges arise mainly from limited temporal frequency and/or spatial resolution of satellite data that can easily miss fast-changing vegetation functions. Therefore, our chosen period is unique in evaluating the efficacy of satellite measurements in capturing such complex dynamics."* **(Line 285-290)**

[RC2-10] Therefore, I recommend a major revision of this manuscript before it can be accepted for publication.

[AR2-10] We sincerely appreciate your constructive comments. We hope our revisions have effectively addressed your concerns and improved the quality of the manuscript.